# Teenage mothers report poor health and economic functioning in Western Kenya: A call to action

Aleksandra Jakubowski[1,2]*, Elizabeth Nakiyingi[1], Jane Wamae[3], Samuel Oyugi[3], Joseph R. Starnes[4], Sandra Mudhune[3], Benson Nyawade[3], Willys Ochieng[5], Erick Kelvin[5], Tom Odhong[5], Ash Rogers[3], Richard Wamai[6,7,8]

1 Department of Public Health and Health Sciences, Northeastern University, Boston, Massachusetts, United States of America, 2 Department of Economics, Northeastern University, Boston, Massachusetts, United States of America, 3 Lwala Community Alliance, Rongo, Kenya, 4 Vanderbilt, University Medical Center, Nashville, Tennessee, United States of America, 5 Department of Health, Migori County Government, Rongo, Kenya, 6 Department of Cultures, Societies, and Global Studies, Northeastern University, Boston, Massachusetts, United States of America, 7 African Centre for Community Investment in Health, Nginyang, Baringo County, Kenya, 8 Integrated Initiative for Global Health, Northeastern University, Boston, Massachusetts, United States of America

* a.jakubowski@northeastern.edu

## Abstract

Teenage pregnancy remains a critical issue in Kenya, with 15% of girls aged 15–19 having been pregnant. Counties in western Kenya experience high teenage pregnancy rates (22–30%) along with high HIV prevalence and widespread poverty. Long-term consequences of teenage pregnancy have been documented in high-income countries, but evidence from the Global South is lacking. Here, we examined the association between teenage pregnancy and adult socio-economic functioning in western Kenya using cross-sectional survey data from Migori County, Kenya. We categorized women into three groups: adult mothers (first child ≥20 years), teenage mothers to 1 child (had 1 child before age 20), and teenage mothers to 2+children (had 2 or more children before age 20). We then compared adult socioeconomic and health outcomes of these groups. We found that among 6,089 mothers, 45.2% had their first child during adolescence. Compared to adult mothers, teenage mothers were significantly less likely to complete primary education: a 12.2 percentage point (pp) reduction (95% CI: -14.9, -9.4) among teenage mothers to 1 child and 27.6 pp reduction (95% CI: -31.4, -23.8) among teenage mothers to 2+children. Teenage mothers were also more likely to have loans and experience food insecurity. The risk of experiencing the death of a child increased from 3.4% among adult mothers to 15.3% among teenage mothers to 2+children, a 4.5-fold increase (p<0.001). Teenage mothers also reported short birth spacing and poor mental health. Our study highlights that teenage motherhood is associated with worse health and economic functioning that persists into adulthood. Urgent policy action is needed to address teenage pregnancy in Kenya to support the economic development of youth and reduce the risk of HIV infection.

**Data availability statement:** De-identified data are available for download from the Harvard Dataverse repository: doi.org/10.7910/DVN/QOPLVI.

**Funding:** The authors received no specific funding for this work.

**Competing interests:** The authors have declared that no competing interests exist.

## Introduction

Teenage pregnancy and childbearing (defined by the World Health Organization (WHO) as a pregnancy or birth occurring to a woman aged 10–19) [1] remains a significant public health problem globally and is the highest in sub-Saharan Africa (SSA) [2]. In 2023, the teenage birth rate was 94 per 1000 women in sub-Saharan Africa compared to 39 per 1000 women globally and 10 per 1000 women in high-income countries [3]. This translates to 12 million babies born to teenage mothers annually, 46% of them in sub-Saharan Africa [2,4,5]. In Kenya, the teenage birth rate is 82 births per 1000 women, though rates vary widely across the country. According to the 2022 Kenya Demographic Health Survey (KDHS), 15.8% of teenage girls have ever been pregnant, with significant within-country variation, ranging from 48.3% of girls in Samburu County to 3.6% of girls in Nyeri County [6]. Counties in the western part of Kenya are home to some of the highest teenage pregnancy and HIV prevalence rates [7]. In 2022, Migori County (the focus of this study) had the fifth highest teenage pregnancy rate in Kenya at 23% among girls aged 15–19 years and the fourth highest HIV prevalence at 12.9% among women aged 15 – 49 years [6,8]. Neighboring counties include Homa Bay with 22.9% teenage pregnancy and 19.8% HIV prevalence, Siaya with 20% teenage pregnancy and 17.5% HIV prevalence, and Kisumu with 11.6% teenage pregnancy and 18.7% HIV prevalence among women aged 15–49 years [6,8]. This part of Kenya also experiences widespread poverty with 48% of the population living in extreme poverty (<$2/day) [9]. Together, these issues pose significant challenges to the development of the youth population in Kenya.

The long-term consequences of teenage pregnancy have been studied extensively in high-income countries. Women who became mothers as teenagers are more likely to be single, poor, and unemployed [10,11], never complete high school or need income assistance [11–13], experience poor self-reported health and depressive symptoms [14], and die prematurely [15]. Studies conducted in low- and middle-income countries (LMICs) primarily focus on the immediate outcomes of becoming a teenage mother and antecedents of teenage pregnancy. This includes studies that highlight the elevated risk of teenage pregnancy complications (e.g., obstructed labor, preterm delivery, low birth weight, and postpartum hemorrhage) [16–18] and qualitative work examining teenage girls' pregnancy experiences such as their school participation, mental health, economic conditions, discrimination, and social norms [19,20]. The key risk factors of teenage pregnancy include poverty, low education, coercive sexual relations, gender norms, peer influence [21–24], alcohol use, low self-esteem [21,25,26], and poor access to contraception and sex education [21,25]. In Kenya, as in much of SSA, research on teenage pregnancy has primarily focused on determinants, immediate outcomes, and the experiences of teenage mothers [17,18,24,27–32]. Child marriage has also been identified as both a risk factor and consequence of teenage pregnancy, deeply rooted in the cultural and gender norms in Kenya [27,33,34]. Additionally, teenage pregnancy is among the leading causes of school dropout among girls in Kenya, significantly altering their educational trajectories and economic opportunities [35,36].

To our knowledge, only two previous studies from LMICs examined long-term consequences of teenage motherhood. In South Africa, researchers found a negative impact on economic and psychological wellbeing but not on physical health outcomes; however, this study did not examine outcomes among women who had multiple children as teens [37]. In Uganda, researchers found that becoming a teenage mother had negative impacts on socioeconomic and reproductive outcomes, and that the effect was magnified by repeated births during teenage years [38]. However, their study restricted the analysis to women aged 40–49 years old, limiting generalizability of their findings. Here, we use a large household survey from Migori County that captures robust economic, physical and mental health measures to test the impact of teenage pregnancy, and the compounded impact of repeated teenage pregnancy, on later life outcomes.

## Methods

### Ethics statement

The household survey's protocol and study design received approval from the African Medical and Research Foundation (AMREF) Health Africa Ethics and Scientific Review Committee on March 29, 2021 (AMREF-ESRC P452/2018), and from the Northeastern University Institutional Review Board on September 21, 2020 (IRB #: 20-09-18). Additionally, a research permit was granted by the National Commission for Science, Technology, and Innovation in Kenya on February 11, 2021 (NACOSTI/P/21/8776). All study staff completed ethical research training, and informed consent was obtained from all participants prior to their involvement in the survey. Both written and verbal consent was obtained from each respondent in the form of a signature or thumbprint and verbal agreement to participate after reading a standardized script informing the respondent of the survey's purpose and confidentiality policy. Respondents who could not read or write, were requested to invite a witness to participate in the consenting process. Minors were not eligible for participation in the LHS survey. Data collection was from 3rd May 2021 until 30th June 2021 [39].

### Overview

We used the 2021 Lwala Household Survey (LHS) to examine the association between teenage motherhood status and adult health and economic functioning. The LHS is a cross-sectional survey of adult respondents (18 years or older) from 7,250 households in Migori County, conducted as part of an ongoing evaluation of the Lwala interventions [39]. Lwala empowers local communities to improve health outcomes through forming community health committees, professionalizing community health promoters, enhancing health facility care, and leveraging data for decision-making [40]. We used the birth history data to identify which respondents became mothers while they were teenagers (ages 10-19) and compared their socio-economic outcomes in adulthood to women who became mothers as adults (ages 20 or later).

### Setting and survey design

Located in the Lake Victoria region of southwestern Kenya, Migori County is a mostly rural area with a population of over 1 million people [41]. Administratively, there are 10 sub-counties, each with multiple sub-divisions called wards [42]. We randomly selected households from 8 wards using a hybrid sampling strategy adapted from the World Health Organization's Expanded Programme on Immunization (EPI). Each ward was divided into 127 grid squares using Geographic Information System (GIS) technology, with enumerators starting each day at a randomly generated central point within each grid. Households were then selected using the "spin-the-bottle" method, in which a random direction was chosen, and households were surveyed sequentially along that path [43]. To reflect the Lwala Community Alliance's programmatic focus on maternal and child health, households with children under five years of age were intentionally oversampled. Specifically, of the seven households surveyed within each grid square, at least five were required to include a child under five. Survey teams prioritized interviewing female heads of household when available, particularly due to modules on family planning and child health. Further details about sampling and survey techniques have been extensively described in prior publications [39].

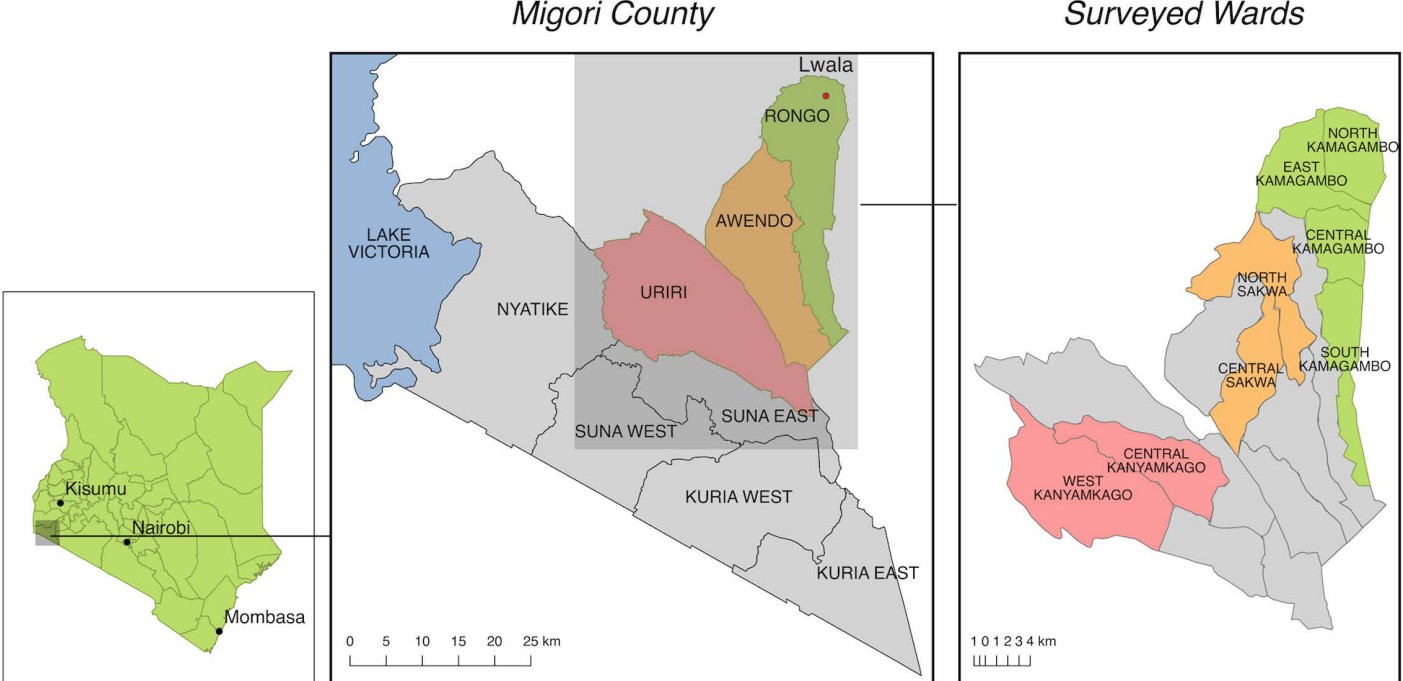

**Fig 1. Map of Kenya and Migori County showing sub-counties where the Lwala Community Alliance works and where the survey has been conducted. Notes:** The map in Fig 1 was created in QGIS using basemap shapefiles available through the Database of Global Administrative Areas. These data are freely available for use in academic publishing (https://gadm.org/license.html). The figure was previously used in other PLOS publications (https://journals.plos.org/plosone/article?id=10.1371/journal.pone.0256555).

## Study groups

Our analytic sample was restricted to female respondents who were 20 years or older and who had given birth to at least one child prior to the survey, resulting in a final sample of 6,089 women. We defined teenage mothers according to the WHO definition as any birth that occurred to a woman ages 10 to 19 years old [1]. We excluded respondents who reported giving birth before age 10 due to the potential for reporting errors and recall bias. Each woman was categorized into a study group: 1) **Adult mothers** - women who had their first child at age 20 or older; **Teenage mothers to 1 child** - women who gave birth to one child before age 20; and **Teenage mothers to 2+ children** - women who had two or more children before age 20.

## Outcome variables

The outcome variables encompass economic, physical and mental health functioning. Economic functioning includes educational attainment (primary school completion), financial insecurity (any loans taken out), and food insecurity (respondent or someone in their household went to sleep hungry in the past four weeks). Physical health functioning includes child mortality (whether any of the respondent's children had died) and short birth spacing (less than two years between births). Mental health was measured using the PHQ-8 depression scale, which identifies eight symptoms of depression and anxiety: 1) little interest or happiness in doing things, 2) feeling down, depressed or hopeless, 3) trouble sleeping or sleeping too much,4) tired or little energy, 5) poor appetite or overeating, 6) felt bad about self or like a failure, 7) trouble concentrating and 8) moved or spoke so slowly that others could notice [44,45]. Respondents were asked to rate whether they had experienced the symptoms in the two weeks preceding the survey on a 4-point scale: 0 = not at all, 1 = several

days, 2 = more than half the days, and 3 = nearly every day. We then calculated a total mental health score for each participant and defined a poor mental health indicator as those women who scored in the top 25th percentile. This was done to identify women with the most severe symptoms within our sample in the absence of a clinical diagnostic cutoff.

## Statistical analyses

Data cleaning and analysis were conducted using Stata version 18 (StataCorp LP, College Station, TX) [46]. We compared the descriptive characteristics of the study participants (age at time of survey, age at first birth, number of children, marital status, religion, household size, wealth index, and birth cohort) across three groups: adult mothers, teenage mothers to 1 child, and teenage mothers to 2 or more children. The wealth index was calculated based on ownership of assets and land, materials used for house construction, and types of water and sanitation facilities. The birth cohort variable was a set of indicators for the decade in which respondents were born. We grouped participants born in the 1950s or earlier into one group because these participants comprised a very small part of the sample (<2%).

We fitted ordinary least squares (OLS) regression models to examine the association between teenage pregnancy and the outcome variables. Each regression model controlled for a range of descriptive characteristics (woman's age, marital status, religion, household size, and wealth index) and included region of residence and birth cohort fixed effects. The region fixed effect controlled for regional variation and the birth cohort fixed effect accounted for temporal trends. The coefficients were multiplied by 100 to represent the percentage point (pp) change from the control mean (reference group are the adult mothers). We set alpha at 0.05 and report the 95% confidence intervals (95% CIs) around the estimates.

## Sensitivity analysis

One of the main variables tested in the paper is child mortality, which is defined as any of the respondents' children having died. The birth and death history data enables us to specify, with some level of uncertainty, whether the child who had died was an infant (<12 months of age) or under 5 years of age at the time of death. We test these alternative outcome variables in sensitivity analysis and interpret the results with caution given the uncertainty in recalling ages at time of death and the small number of women in the sample reporting infant deaths.

We checked whether the results were robust to alternative model specifications by fitting logit and probit models. We also made sure that including the eldest participants did not drive the study findings by excluding any participant born before the 1980s from the sample. Lastly, we tested whether restricting the sample to participants who had non-missing data for all variables included in the models affected the results.

## Results

### Demographics

The original LHS sample of 7,250 households was restricted to 6,089 female respondents who had given birth to at least one child and were 20 years or older at time of survey (Fig A in S1 Appendix). Our final analytic sample included 3,230 (54.9%) adult mothers (first birth at age 20 or older), 1,813 (30.8%) teenage mothers to 1 child, and 845 (14.4%) teenage mothers to 2 + children.

The average age of women in the sample was 30.1 years (Table 1), with adult mothers slightly older than teenage mothers to 1 child (31.5 years vs. 28.1 years, respectively). Adult mothers had their first child at average age of 23.5 years compared to teenage mothers to 1 child at age 17.7 years and teenage mothers to 2 + children at age 15.3 years. The average number of children born to adult mothers was 2.1 children compared to 2.7 children for teenage mothers to 1 child and 4.1 children for teenage mothers to 2 + children. Most study participants were married across all study groups (83.5% of adult mothers and 89.5% of teenage mothers). Teenage mothers to 2 + children lived in larger households compared to the other groups. We found a clear wealth gradient among the three study groups: adult mothers were much more likely to live in the wealthy households (23.7% lived in the wealthiest households vs. 19.5%

**Table 1. Descriptive characteristics of study sample.**

| | Adult mothers | Teenage mothers, 1 child | Teenage mothers, 2+children | Total |
|---|---|---|---|---|
| Number of participants | 3,230 (54.9%) | 1,813 (30.8%) | 845 (14.4%) | 6,089 |
| Age in years, x̄ (SD) | 31.5 (9.3) | 28.1 (7.8) | 29.5 (8.2) | 30.1 (9.0) |
| Woman's age at first birth, x̄ (SD) | 23.5 (4.2) | 17.7 (1.5) | 15.3 (1.7) | 20.5 (4.7) |
| Number of children, x̄ (SD) | 2.1 (1.4) | 2.7 (1.6) | 4.1 (1.7) | 2.5 (1.7) |
| Married or cohabiting | 2,696 (83.5%) | 1,622 (89.5%) | 757 (89.6%) | 5,249 (86.2%) |
| Religion | | | | |
| None | 8 (0.3%) | 2 (0.1%) | 2 (0.2%) | 12 (0.2%) |
| Catholic | 506 (15.9%) | 271 (15.2%) | 113 (13.7%) | 941 (15.7%) |
| Seventh Day Adventist | 1,457 (45.7%) | 698 (39.1%) | 271 (32.9%) | 2,505 (41.9%) |
| Protestant | 658 (20.6%) | 383 (21.5%) | 185 (22.5%) | 1,231 (20.6%) |
| Roho Church | 435 (13.6%) | 336 (18.8%) | 207 (25.2%) | 1,014 (17.0%) |
| Legio Maria | 34 (1.1%) | 31 (1.7%) | 18 (2.2%) | 85 (1.4%) |
| African Independent Church | 84 (2.6%) | 61 (3.4%) | 27 (3.3%) | 180 (3.0%) |
| Islam | 7 (0.2%) | 3 (0.2%) | 0 (0.0%) | 10 (0.2%) |
| Household size, x̄ (SD) | 3.9 (1.5) | 4.5 (1.6) | 5.5 (1.6) | 4.3 (1.6) |
| Wealth Index (in quintiles) | | | | |
| Poorest | 625 (19.5%) | 375 (20.8%) | 191 (22.7%) | 1,208 (20.0%) |
| Poor | 552 (17.2%) | 376 (20.8%) | 190 (22.5%) | 1,196 (19.8%) |
| Middle | 581 (18.1%) | 381 (21.1%) | 211 (25.0%) | 1,208 (20.0%) |
| Wealthy | 690 (21.5%) | 346 (19.2%) | 135 (16.0%) | 1,216 (20.1%) |
| Wealthiest | 759 (23.7%) | 327 (18.1%) | 116 (13.8%) | 1,225 (20.2%) |
| Birth cohort | | | | |
| 1930s, 1940s & 1950s | 62 (1.9%) | 12 (0.7%) | 6 (0.7%) | 89 (1.5%) |
| 1960s | 86 (2.7%) | 24 (1.3%) | 25 (3.0%) | 139 (2.3%) |
| 1970s | 250 (7.7%) | 76 (4.2%) | 37 (4.4%) | 368 (6.0%) |
| 1980s | 921 (28.5%) | 386 (21.3%) | 228 (27.0%) | 1563 (25.7%) |
| 1990s | 1,782 (55.2%) | 1,043 (57.5%) | 472 (55.9%) | 3,436 (56.4%) |
| 2000s | 129 (4.0%) | 272 (15.0%) | 77 (9.1%) | 493 (8.1%) |

**Notes:** Descriptive characteristics of study sample. All variables presented here were included in the regression models as covariates. **Abbreviations:** N – number, x̄ – mean, SD – standard deviation, % – percent

in the poorest households), and teenage mothers to 2+children were concentrated in the less wealthy households (22.7% in the poorest households vs. 13.8% in the wealthiest households). Most participants were born between 1980 and 1999.

The unadjusted mean outcomes show large differences between the three study groups (Table 2). Among adult mothers in the sample, 53.7% completed primary education compared to 42.1% of teenage mothers to 1 child and 18.6% of teenage mothers to 2+children. Adult mothers were less likely to take out loans or experience food insecurity compared to teenage mothers. The proportion of adult mothers who had experienced child mortality was 3.3% compared to 7.0% among teenage mothers to 1 child and 13.0% among teenage mothers to 2+children. Adult mothers were least likely to have poor mental health (28.6%) compared to teenage mothers to 1 child (32.9%) and teenage mothers to 2+children (40.4%). Women who became mothers as teens were more likely to have short birth spacing: 75% of teenage mothers to 2+children compared to 30% of teenage mothers to 1 child and 27% of adult mothers.

**Table 2. Health and economic outcomes by study groups.**

| | Study groups | | | p-values | | |
|---|---|---|---|---|---|---|
| | **(1) Adult mother** | **(2) Teenage mother, 1 child** | **(3) Teenage mother, 2+children** | Test: (1) = (2) | Test: (1) = (3) | Test: (2) = (3) |
| Completed primary education | 1736 (53.7%) | 763 (42.1%) | 157 (18.6%) | <0.001 | <0.001 | <0.001 |
| Has loans | 591 (18.9%) | 392 (22.1%) | 238 (28.6%) | 0.001 | <0.001 | 0.0542 |
| Food insecurity | 266 (8.3%) | 201 (11.1%) | 170 (20.2%) | 0.001 | <0.001 | <0.001 |
| Child mortality | 107 (3.3%) | 127 (7.0%) | 110 (13.0%) | <0.001 | <0.001 | <0.001 |
| Poorest mental health | 924 (28.6%) | 596 (32.9%) | 341 (40.4%) | 0.011 | <0.001 | <0.001 |
| Short birth spacing | 874 (27.1%) | 548 (30.2%) | 633 (74.9%) | 0.004 | <0.001 | 0.0113 |

**Notes:** Unadjusted summary statistics of main study outcomes. *p*-values are from t-tests comparing the differences between study groups: (1) adult mothers (2) teenage mothers to 1 child, (3) teenage mothers to 2+children. **Abbreviations:** %, percent

## Socioeconomic functioning

We found statistically significant and meaningful differences in economic functioning among the respondents based on the age at which they first became mothers. Compared to adult mothers (the reference group), teenage mothers to 1 child were 12.2 pp (95% CI: -14.9, -9.4) less likely to finish primary school, while teenage mothers to 2+children were 27.6 pp (95% CI: -31.4, -23.8) less likely to complete primary school ([Fig 2]). The likelihood of taking out loans was higher among teenage mothers: 4.3 pp (95% CI: 1.8, 6.8) higher for those with 1 child and 7.7 pp (95% CI: 4.2, 11.2) higher for those with 2+children, relative to adult mothers. Similarly, food insecurity in the past four weeks was more prevalent among teenage mothers, with increases of 3.1 pp (95% CI: 1.2, 5.0) and 9.2 pp (95% CI: 6.6, 11.8) among those with 1 child and 2+children, respectively, relative to adult mothers.

## Health outcomes

Becoming a mother as a teenager was significantly associated with negative health outcomes, especially among women who had multiple children during adolescence. Teenage mothers to 2+children were 11.9 pp (95% CI: 9.9, 13.9) more likely to experience child mortality compared to adult mothers ([Fig 3]). In absolute terms, 3.4% of adult mothers experienced the death of a child compared to 15.3% of teenage mothers to 2+children, representing a 4.5-fold increase in risk. Short birth spacing was more prevalent among teenage mothers to 2+children compared to the other two study groups. Teenage mothers with 2+children were 24.9 pp (95% CI: 21.6, 28.2) more likely to have short birth spacing compared to adult mothers. There was no meaningful difference in birth spacing between adult mothers and teenage mothers to 1 child.

## Mental health

Adult women who became mothers as teens reported poor mental health wellbeing. Using an indicator for women who scored in the highest quartile of the PHQ-8 (top 25th percentile), indicating worse mental health, we found that teenage mothers to 1 child were 4.2 pp (95% CI: 1.3, 7.1) more likely to have poor mental health and teenage mothers to 2+children were 9.2 pp (95% CI: 5.2,13.2) more likely to have poor mental health compared with adult mothers ([Fig 3]).

When we examined the components of the PHQ-8 we found that teenage mothers to 2+children were significantly more likely to disclose feeling each of the symptoms we measured ([Fig 4]). In many cases, teenage mothers to 2+children had significantly worse mental health functioning even when compared to teenage mothers to 1 child. The differences between adult mothers and teenage mothers to 1 child were less pronounced and often not statistically significant.

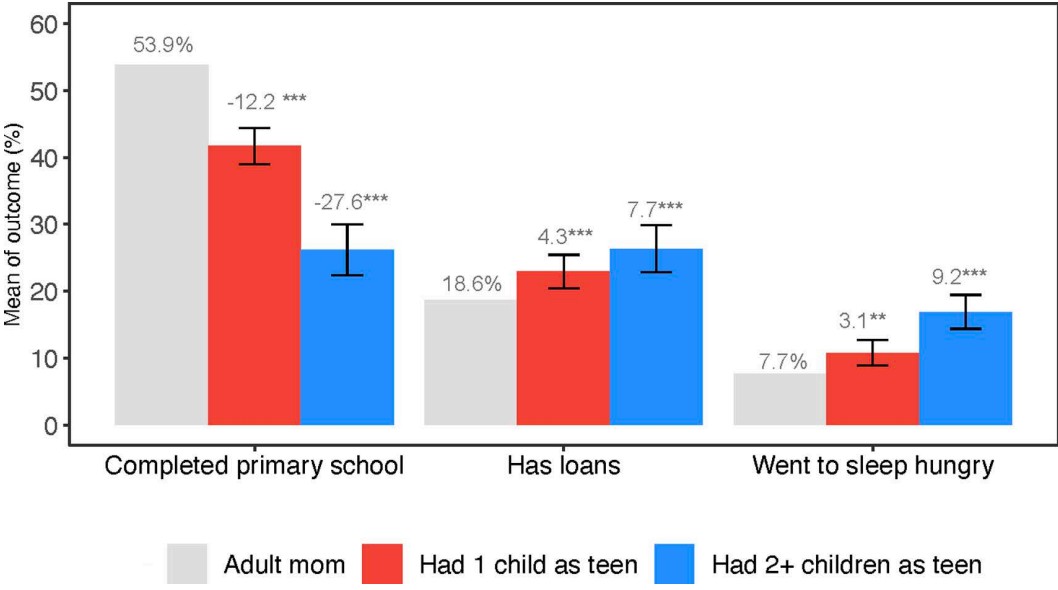

**Fig 2. Teenage mothers reported significantly less favorable economic outcomes than adult mothers. Notes:** Gray bars represent the control group mean (*adult mothers*). Red bars represent the control group mean plus coefficient on *teenage mothers to 1 child*. Blue bars represent the control group mean plus coefficient on *teenage mothers to 2+children*. The coefficients, which can be interpreted as pp difference from the control mean, are reported as βs above the bars with *p*-value notation: *** p<0.001, ** p<0.01, * p<0.05. The error bars are 95% CIs around the coefficients. The outcomes are labeled below bars along the *x*-axis. Each outcome was estimated using a separate regression model that controlled age, age-squared, marital status, religion, household size, wealth index, region of residence, and birth cohort fixed effect. Regression results are also presented in Table A in S1 Appendix.

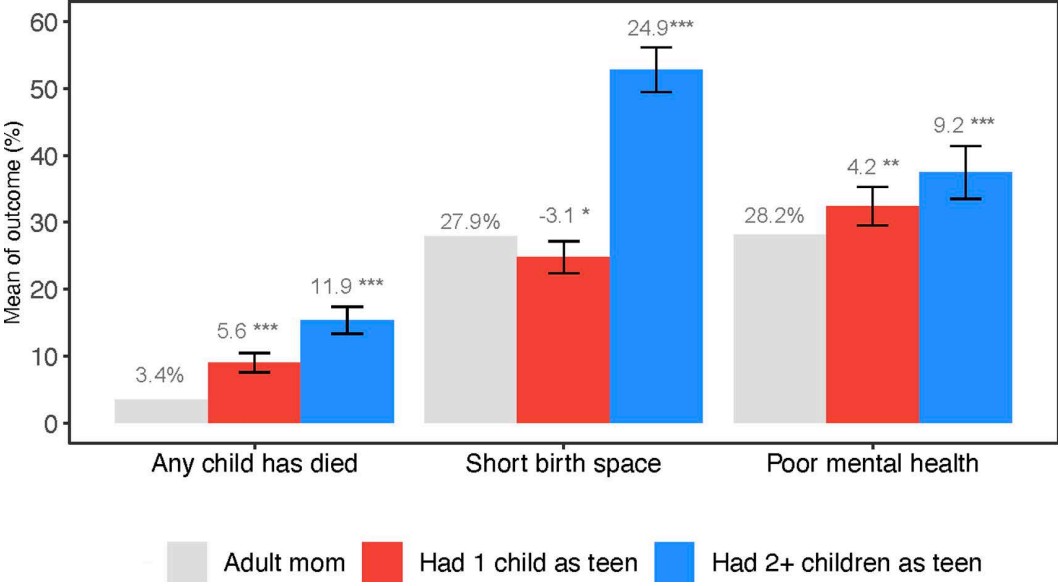

**Fig 3. Teenage mothers reported significantly higher mortality of their children, short birth spacing, and poor mental health. Notes:** Gray bars represent the control group mean (*adult mothers*). Red bars represent the control group mean plus coefficient on *teenage mothers to 1 child*. Blue bars represent the control group mean plus coefficient on *teenage mothers to 2+children*. The coefficients, which can be interpreted as pp difference from the control mean, are also reported as βs above the bars. The error bars are 95% CIs around the coefficients. The outcomes are labeled below bars along the x-axis. Each outcome was estimated using a separate regression model that controlled age, age-squared, marital status, religion, household size, wealth index, region of residence, and birth cohort fixed effect. Regression results are also presented in Table A in S1 Appendix.

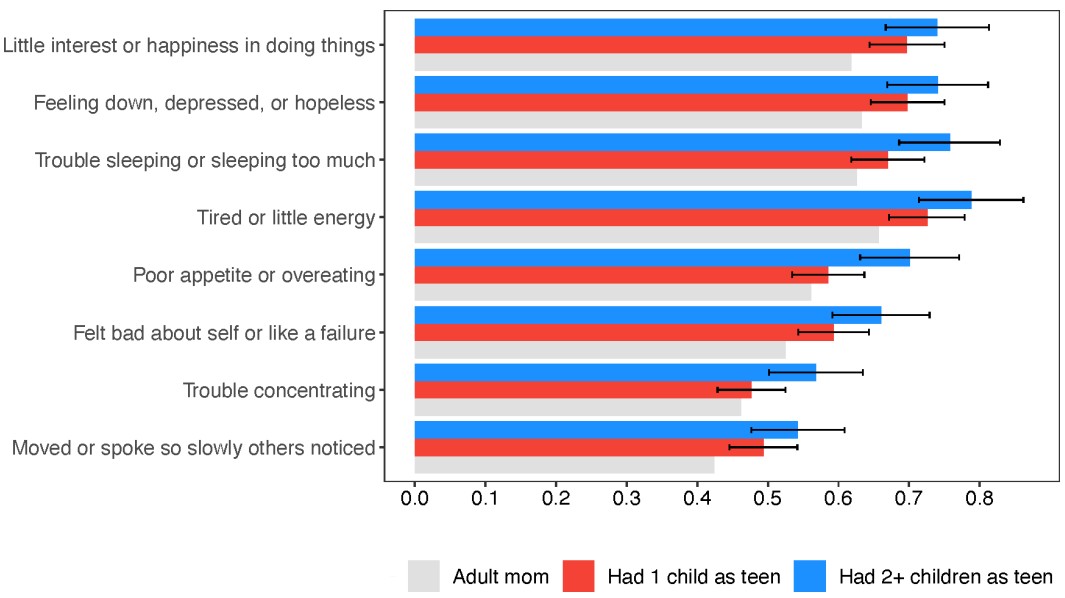

**Fig 4. Teenage mothers to 2＋children scored significantly higher on the PHQ-8 scale. Notes:** Gray bars represent the control group mean (*adult mothers*). Red bars represent the control group mean plus coefficient on *teenage mothers to 1 child*. Blue bars represent the control group mean plus coefficient on *teenage mothers to 2＋children*. The coefficients can be interpreted as the difference from the control mean on a 4-point scale, where 0＝not at all, 1＝several days, 2＝more than half the days, and 3＝nearly every day. Higher score indicates worse mental health functioning. The error bars are 95% CIs around the coefficients. The outcomes are labeled next to bars along the y-axis. Each outcome was estimated using a separate regression model that controlled age, age-squared, marital status, religion, household size, wealth index, region of residence, and birth cohort fixed effect. Regression results are also presented in Table B in S1 Appendix.

## Sensitivity analysis

We found no statistically significant differences in infant mortality (deaths under 12 months) between the study groups (Table C in S1 Appendix). Teenage mothers were significantly more likely to report the death of a child under 5 years old. Compared with adult mothers, teenage mothers to 1 child were 0.7 pp (95% CI: 0.1, 1.2) more likely to have lost a child under 5 and teenage mothers to 2＋children were 1.3 pp (95% CI: 0.6, 2.01) more likely to have experienced this outcome. Results were robust to alternative model specifications in terms of magnitude and statistical significance (Table D in S1 Appendix and Table E in S1 Appendix), to the exclusion of the eldest participants from the model (Table F in S1 Appendix), and to restricting the sample to participants with non-missing data on all variables (Table G in S1 Appendix).

## Discussion

We investigated the socioeconomic and health outcomes of women in Migori County, located in western Kenya, using a large household survey conducted in 2021. We found that teenage mothers, particularly those who had multiple births during adolescence, reported meaningful and statistically significant reductions in socio-economic functioning, physical health, and mental wellbeing in adulthood. These results are consistent with existing evidence from high-income countries [10–15] and two prior studies from LMICs [37,38]. We found that becoming a teenage mother halves a woman's likelihood of completing primary education, and that this has negative repercussions on women's economic functioning in adulthood [13,37,38]. We extend the literature on this topic by adding more robust economic and health outcomes and demonstrating that teenage pregnancy can trigger vicious cycles of poverty and financial instability [12,37,38]. Our results highlight the compounded effects of having more than one child during this critical time of development. The interruption to schooling and skill acquisition disadvantages young women, diminishing their ability to meet basic household needs such as food and

exacerbating the risk of hunger [12,47,48]. Consequently, teenage mothers rely more heavily on loans, often trapping them in poverty cycles. The economic hardships faced by teenage mothers, particularly those with multiple children, are profound and require interventions that focus on economic empowerment to improve their long-term wellbeing [49,50].

Among the study's most striking results, we found that women who had multiple children as teenagers were over three times more likely to experience the death of a child compared to those women who first gave birth as adults [38,51]. Extensive literature documents that adolescent mothers face increased risk of pregnancy and birth complications such as obstructed labor, preterm delivery, and postpartum hemorrhage [16–18]. Our study identifies an additional factor contributing to higher child mortality rates among teenage mothers: short birth spacing, which adversely affects both maternal and child health [52,53]. Women who had 2 + children as teenagers were 27 pp more likely (over three times more likely in absolute terms) to have short birth intervals of less than 2 years between their children, a well-established risk factor for child mortality [52,53]. This increased risk may stem from limited access to contraception and postpartum family planning counseling, stigma from healthcare providers, and societal pressures to have more children [54,55]. Teenage mothers may also face barriers such as low autonomy in reproductive decision making and limited knowledge about birth spacing and contraception [56,57]. These findings underscore the urgent need for targeted postpartum family planning programs for teenage mothers and adolescent-friendly reproductive health services to address these barriers [58].

Another key finding is the marked decline in mental health among teenage mothers. Previous research has documented the isolation and stigma surrounding teenage pregnancy [20,30,59,60], factors that contribute to poor mental health outcomes. Our study found that these mental health challenges persist into adulthood, underscoring the critical importance of addressing mental wellbeing of teenage mothers. Among the eight PHQ-8 symptoms we measured, teenage mothers to 2 + children scored significantly higher on each item, including having little interest in things, feeling bad about oneself or like a failure, having trouble sleeping, concentrating, and feeling tired. The severity of poor health and socio-economic outcomes we observed likely contributed to this mental health burden. Our findings are in line with one other study that measured long term mental health of teenage mothers in South Africa [37] and with similar research in high income countries [14]. Poor mental health among teenage mothers can impair parenting capacity and reduce labor force participation, worsening economic outcomes [61,62]. Additionally, depressive symptoms are associated with lower health-seeking behavior, which can compound physical health issues for mothers and children [63].

We found significant differences in fertility among the study groups: teenage mothers to 2 + children had an average of four children, compared to three children among teenage mothers to 1 child, and two children among adult mothers. This pattern is consistent with findings from Amongin et al.'s study in Uganda [38]. Notably, because our sample includes women of childbearing age whose fertility is ongoing and the teenage mother groups are younger than adult mothers, our results likely underestimate fertility among teenage mothers in Kenya.

Our study found that 46.9% of the sampled women were teenage mothers, a prevalence substantially higher than the 23% reported for Migori County in the KDHS [6]. This discrepancy likely arises from differences in population definitions: our retrospective analysis was based on a sample of women aged 20–85 years old at the time of the survey, whereas the KDHS calculates teenage birth prevalence among girls aged 15–19 years. Amongin et al.'s study similarly found a 36.2% teenage motherhood prevalence in their national sample of Ugandan women aged 40–49 years. The higher prevalence we found likely stems from our study's focus on rural and less wealthy respondents from a region that is highly affected by HIV and teenage pregnancy.

Overall, teenage mothers, especially those who had multiple children, face substantially worse socio-economic outcomes as adults than women who first gave birth in adulthood. These outcomes likely reflect both causes and consequences of early childbearing. Low socioeconomic status is a known risk factor for teenage pregnancy, as impoverished girls often lack access to quality education and reproductive health services [21,64]. Becoming a mother during adolescence can exacerbate these hardships, especially when girls drop out of school, which reduces future earning potential and increases the need for informal borrowing or high-interest loans [12,65,66]. Limited earning capacity also raises the

risk of food insecurity, which further harms maternal and child health outcomes [67–69]. These adverse socioeconomic conditions trap mothers in debt cycles by reducing their financial independence [65], limiting access to healthcare (which increases maternal complications, poor birth outcomes, and child mortality) [70], and exacerbating psychological distress such as feelings of hopelessness, depression and anxiety (which further deepens financial insecurity). Children of teenage mothers who grow up in poverty are also more likely to experience poor health, drop out of school, need income assistance, and repeat cycles of early pregnancy and poverty [12,71,72].

Our study documents the detrimental impact of teenage pregnancy on education, economic functioning, and fertility decisions such as birth spacing and parity, and mental health. These findings are especially relevant in the context of western Kenya, an area with some of the world's highest HIV prevalence [7,8]. Existing literature from LMICs, including Kenya, largely relies on the nationally representative DHS data to explore teenage pregnancy, which provide limited samples from specific geographical areas. For instance, although Migori County is home to some of Kenya's highest HIV prevalence and teenage pregnancy, the DHS only sampled 777 respondents from this county, 569 of whom were mothers [6]. Our larger sample allowed for more nuanced analyses by dividing teenage mothers into two groups based on parity to assess the differential effect of multiple adolescent births.

Our study had multiple limitations. The results can only be interpreted as associations given the cross-sectional nature of the data and the endogeneity of teenage motherhood and economic functioning. We did not have sufficient sample size to estimate the association between teenage motherhood and infant mortality with precision. The child mortality outcomes relied on self-reported birth and death histories, which may be subject to recall bias, especially for events that occurred many years prior to the survey. This may introduce uncertainty in distinguishing between infant and under-five mortality. The study used retrospective analysis techniques to assign women into study groups based on when they first became mothers. It is possible that recalling the years in which respondents' children were born includes some recall bias, but we do not expect this to meaningfully affect our results. Finally, the rural, high HIV prevalence, and impoverished context of Migori County may limit the generalizability of results to wealthier, urban, or low HIV prevalence settings.

Addressing teenage pregnancy in western Kenya requires targeted, multisectoral interventions that address both its root causes and long-term consequences. Programs integrating health interventions with financial literacy, economic opportunity, and social support, particularly those aimed at reducing borrowing and promote financial stability, may be particularly effective. Kenya is already prioritizing youth-focused investments through national initiatives such as the Kenya Youth Employment and Opportunities Project [73], the Return to School policy [74], and the Adolescent Sexual and Reproductive Health Policy. These programs align with national goals to eliminate teenage pregnancy by 2030 [75] and empower women [76]. At the county level, Migori has identified teenage pregnancy as a policy priority in its Multi-Sectoral Action Plan for Adolescents and Youth, encouraging collaboration across health, education, and youth sectors [77]. The Lwala Community Alliance also runs programs to improve the economic wellbeing of youth affected by teenage pregnancy, including Broadened Horizon, which empowers teenage mothers to return to school [40]. Our study underlines the urgent need to also address the mental toll of teenage pregnancy. The persistent disadvantages young mothers face, including limited education, economic opportunity, food security, and child loss, take a tremendous mental health toll that extends into adulthood.

Preventing teenage pregnancy and supporting those who become mothers as adolescents is both a public health emergency and an economic development priority for Kenya, and is essential to interrupting cycles of poverty and inequality. Our findings emphasize the critical need for coordinated multisectoral action to promote long term wellbeing of teenage mothers in Kenya.

## Supporting information

**S1 Appendix. Supplementary tables and figures. Fig A.** Participant flow chart showing the original sample of LHS respondents that includes both men and women (N = 7,250) and how each study eligibility criteria affected sample size. **Table A.** Regression results for education and health outcomes presented in table format. **Table B.** Regression results

for mental health outcomes presented in table format. **Table C.** Regression models using alternative definitions of child mortality as the outcome variable. **Table D.** Sensitivity analysis: logistic regression models with marginal effects presented in table. **Table E.** Sensitivity analysis: probit regression models with marginal effects presented in table. **Table F.** Sensitivity analysis: regression model excluding older participants (born before 1980s). **Table G.** Sensitivity analysis: regression model restricting sample to participants with non-missing data for each outcome variable.
(PDF)

**S1 Checklist.  Inclusivity in global research.**
(DOCX)

## Acknowledgments

We would like to acknowledge the Lwala Community Health Alliance for their ongoing work in Migori County and the effort in collecting the survey and managing the data. We would like to thank the study respondents for their time and participation in the survey.

## Author contributions

**Conceptualization:** Aleksandra Jakubowski.

**Data curation:** Jane Wamae, Joseph R. Starnes.

**Formal analysis:** Aleksandra Jakubowski.

**Investigation:** Jane Wamae, Sam Oyugi, Joseph R. Starnes, Sandra Mudhune.

**Methodology:** Aleksandra Jakubowski.

**Project administration:** Jane Wamae, Sam Oyugi, Sandra Mudhune.

**Software:** Aleksandra Jakubowski.

**Supervision:** Sandra Mudhune, Benson Nyawade, Tom Odhong, Ash Rogers, Richard Wamai.

**Visualization:** Aleksandra Jakubowski.

**Writing – original draft:** Aleksandra Jakubowski, Elizabeth Nakiyingi, Jane Wamae, Joseph R. Starnes, Tom Odhong, Richard Wamai.

**Writing – review & editing:** Aleksandra Jakubowski, Elizabeth Nakiyingi, Jane Wamae, Sam Oyugi, Joseph R. Starnes, Sandra Mudhune, Benson Nyawade, Willys Ochieng, Erick Kelvin, Tom Odhong, Ash Rogers, Richard Wamai.

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
