## [Decision Letter · Decision Letter 0]

PGPH-D-25-00394

Teen Mothers Report Poor Health and Economic Functioning in Western Kenya: A call to action

Dear Authors,

Thank you for submitting your manuscript to PLOS Global Public Health. After careful consideration, we feel that it has merit but does not fully meet PLOS Global Public Health’s publication criteria as it currently stands. Therefore, we invite you to submit a revised version of the manuscript that addresses the points raised during the review process.

We look forward to receiving your revised manuscript.

Kind regards,

Kavitha Ranganathan, MD

Academic Editor

Journal Requirements:

 1. Please include a complete copy of PLOS’ questionnaire on inclusivity in global research in your revised manuscript. Our policy for research in this area aims to improve transparency in the reporting of research performed outside of researchers’ own country or community. The policy applies to researchers who have travelled to a different country to conduct research, research with Indigenous populations or their lands, and research on cultural artefacts. The questionnaire can also be requested at the journal’s discretion for any other submissions, even if these conditions are not met. Please find more information on the policy and a link to download a blank copy of the questionnaire here: https://journals.plos.org/globalpublichealth/s/best-practices-in-research-reporting. Please upload a completed version of your questionnaire as Supporting Information when you resubmit your manuscript.  2. Please insert an Ethics Statement at the beginning of your Methods section, under a subheading 'Ethics Statement'.  3. Please provide separate figure files in .tif or .eps format. For more information about figure files please see our guidelines:   https://journals.plos.org/globalpublichealth/s/figures https://journals.plos.org/globalpublichealth/s/figures#loc-file-requirements   4. We have noticed that you have cited Table 3 in the manuscript file but there are no corresponding tables in the manuscript. Please amend your manuscript to include this table, noting that tables should not be uploaded as individual files.  5. We have noticed that you have uploaded Supporting Information files, but you have not included a list of legends. Please add a full list of legends for your Supporting Information files after the references list.  6. In the online submission form, you indicated that “De-identified data will be made available upon request.”.  All PLOS journals now require all data underlying the findings described in their manuscript to be freely available to other researchers, either 1. In a public repository, 2. Within the manuscript itself, or 3. Uploaded as supplementary information. This policy applies to all data except where public deposition would breach compliance with the protocol approved by your research ethics board. If your data cannot be made publicly available for ethical or legal reasons (e.g., public availability would compromise patient privacy), please explain your reasons by return email and your exemption request will be escalated to the editor for approval. Your exemption request will be handled independently and will not hold up the peer review process, but will need to be resolved should your manuscript be accepted for publication. One of the Editorial team will then be in touch if there are any issues.  7. Some material included in your submission may be copyrighted. According to PLOS’s copyright policy, authors who use figures or other material (e.g., graphics, clipart, maps) from another author or copyright holder must demonstrate or obtain permission to publish this material under the Creative Commons Attribution 4.0 International (CC BY 4.0) License used by PLOS journals. Please closely review the details of PLOS’s copyright requirements here: PLOS Licenses and Copyright. If you need to request permissions from a copyright holder, you may use PLOS's Copyright Content Permission form. Please respond directly to this email or email the journal office and provide any known details concerning your material's license terms and permissions required for reuse, even if you have not yet obtained copyright permissions or are unsure of your material's copyright compatibility.  Potential Copyright Issues: a. Figure 1:  please (a) provide a direct link to the base layer of the map (i.e., the country or region border shape) and ensure this is also included in the figure legend; and (b) provide a link to the terms of use / license information for the base layer image or shapefile. We cannot publish proprietary or copyrighted maps (e.g. Google Maps, Mapquest) and the terms of use for your map base layer must be compatible with our CC-BY 4.0 license.  Note: if you created the map in a software program like R or ArcGIS, please locate and indicate the source of the basemap shapefile onto which data has been plotted. If your map was obtained from a copyrighted source please amend the figure so that the base map used is from an openly available source. Alternatively, please provide explicit written permission from the copyright holder granting you the right to publish the material under our CC-BY 4.0 license. Please note that the following CC BY licenses are compatible with PLOS license: CC BY 4.0, CC BY 2.0 and CC BY 3.0, meanwhile such licenses as CC BY-ND 3.0 and others are not compatible due to additional restrictions.  If you are unsure whether you can use a map or not, please do reach out and we will be able to help you. The following websites are good examples of where you can source open access or public domain maps: * U.S. Geological Survey (USGS) - All maps are in the public domain. (http://www.usgs.gov) * PlaniGlobe - All maps are published under a Creative Commons license so please cite “PlaniGlobe, http://www.planiglobe.com, CC BY 2.0” in the image credit after the caption. (http://www.planiglobe.com/?lang=enl) * Natural Earth - All maps are public domain. (http://www.naturalearthdata.com/about/terms-of-use/)

Additional Editor Comments (if provided):

Reviewers' comments:

Reviewer's Responses to Questions

**Comments to the Author**

1. Does this manuscript meet PLOS Global Public Health’s publication criteria ? Is the manuscript technically sound, and do the data support the conclusions? The manuscript must describe methodologically and ethically rigorous research with conclusions that are appropriately drawn based on the data presented.

Reviewer #1: Yes

Reviewer #2: Yes

2. Has the statistical analysis been performed appropriately and rigorously?

Reviewer #1: Yes

Reviewer #2: Yes

3. Have the authors made all data underlying the findings in their manuscript fully available (please refer to the Data Availability Statement at the start of the manuscript PDF file)?

Reviewer #1: No

Reviewer #2: Yes

4. Is the manuscript presented in an intelligible fashion and written in standard English?

Reviewer #1: No

Reviewer #2: Yes

5. Review Comments to the Author

Reviewer #1: Thank you so much for the opportunity to review this very meaningful paper, studying a previously underexplored area. While the content is incredibly insightful the scientific writing of this paper could be very much improved.

Overall:

1. Please correct grammatical errors throughout the piece, some have been pointed others have not. Would strongly suggest reading through the whole piece and improving writing, grammar and sentence structure to improve readability – especially in the introduction and discussion

Abstract:

1. ‘Teenage pregnancy remains a critical issue in Kenya, with 15% of girls 15-19 having been pregnant.’ Please fix the grammar of this sentence, e.g., ‘Teenage pregnancy remains a critical issue in Kenya, with 15% of girls aged 15-19 having been pregnant.

Intro:

1. The paper might benefit with a clear statement upfront defining teen pregnancy. E.g. Teen pregnancy is defined as pregnancy from the ages of 15-19… (ref) This is defined in the methods but as is it is the premise of the paper an upfront definition is always helpful.

2. In 2022, Migori County (the focus of this study) had the fifth highest teen pregnancy in Kenya at 23% of women 15-19 years and adult (15-49) HIV prevalence of 9.73%.

Please clarify ages counted as adult for HIV prevalence? A teenage pregnancy is 15-19 years of age, however adult HIV prevalence counts as 15 years old or onwards?

Methods:

1. These sampling and survey techniques have been extensively described in prior studies.(38) Briefly, we implemented the World Health Organization (WHO) Expanded Programme of Immunization (EPI) method(40) to randomly select the sampling frame, and oversampled households that included women with children under 5 years.(38)

Would suggest flipping these sentences around giving a clear explanation of methodology for sampling and then mentioning extensive description elsewhere.

2. Would an eleven-year-old giving birth classify as a teen mother to a child? Is there a lower cut off age?

3. Is there a justification of the 25th percentile mental health cut off or is it arbitrary?

4. ‘The region fixed effect control for regional variation and birth cohort fixed effect accounted for temporal trends.’ Fix grammar please.

5. The uncertainty surrounding child mortality alongside recall bias is not something that should be reported in the methodology but in the limitations section under discussion. What was conducted is what must be relayed in the methodology.

6. Does AMREF have a full form?

Results:

1. ‘…1,813 (30.8%) Teen mothers’ The T does not need to be capitalized.

2. For results reported in the text have n and % from the table in the text as well, e.g. ‘Majority of the participants were married across all groups, and most were Seventh Day Adventist. (45.7%, n=1,457)’ Repeat as needed where needed.

3. ‘We found a clear wealth gradient among the three study groups: adult mothers were much more likely to live in the wealthiest households, and teen mothers to 2+ children were more concentrated in the least wealthy households. For example, 23.7% of adult mothers lived in the wealthiest households compared to 13.8% of teen mothers to 2+ children.’ The second sentence is not an example of the first it is the verification of it, both sentences could be combined or at the very least the phrase for example could be removed.

4. Remove doubling and quadrupling.

5. Keep terms like food insecurity, short birth spacing and child mortality consistent in the results (table, text and figures) as explained in the methods, no need to revert back to slept hungry or death of a child.

Discussion:

1. Would not describe findings as ‘striking’ consistently.

2. Perhaps elaborate on the reasons for short birth spacing? To identify areas of intervention?

3. Instead of describing poor mental health what are the repercussions of the same?

4. Need to elaborate on the impact of poor socioeconomic outcomes (increased loans, food insecurity) amongst this group and possible interplay with teenage pregnancy or socioeconomic status and what this could mean?

5. ‘We had sufficient sample size to explore the topic of teenage pregnancy with greater nuance by dividing the teen mothers into two groups and finding the differential effect of having multiple children as teenager.’ The grammar of this sentence needs to be fixed. Sample size information and adequacy should be in the methodology.

6. Is it possible to elaborate upon and contextualize the recommendations based on the findings and local need/possibility in Kenya.

Reviewer #2: I commend the authors for a strong research study answering an important question based on available data, building on prior research study findings and adding important additional insights to the work and field. The study is well conceived, well written, and well presented. My single critique is that the abstract says “ Our study highlights that teenage motherhood has negative consequences on health and economic functioning that persist until adulthood” when, of course, the authors are well aware that the consequences cannot be so clearly defined and that these are associations (the authors go to great extent in the discussion to be clear about this so I assume this is a typo/missed detail and recommend it is corrected with something along the lines of “Our study highlights…teenage motherhood IS ASSOCIATED WITH negative consequences…” I have no other specific critiques and recommend the study is accepted with the minor aforementioned change.

6. PLOS authors have the option to publish the peer review history of their article (what does this mean? ). If published, this will include your full peer review and any attached files.

**Do you want your identity to be public for this peer review?** For information about this choice, including consent withdrawal, please see our Privacy Policy .

Reviewer #1: No

Reviewer #2: No

---

## [Editor Report · Decision Letter 1]

Teen Mothers Report Poor Health and Economic Functioning in Western Kenya: A Call to Action

PGPH-D-25-00394R1

Dear Dr. Jakubowski,

We are pleased to inform you that your manuscript 'Teen Mothers Report Poor Health and Economic Functioning in Western Kenya: A Call to Action' has been provisionally accepted for publication in PLOS Global Public Health.

Best regards,

Kavitha Ranganathan, MD

Academic Editor